# Notch Partners in the Long Journey of T-ALL Pathogenesis

**DOI:** 10.3390/ijms24021383

**Published:** 2023-01-10

**Authors:** María Luisa Toribio, Sara González-García

**Affiliations:** Immune System Development and Function Unit, Centro de Biología Molecular Severo Ochoa, Consejo Superior de Investigaciones Científicas (CSIC), Universidad Autónoma de Madrid (UAM), 28049 Madrid, Spain

**Keywords:** NOTCH, T-cell acute lymphoblastic leukemia (T-ALL), thymus, T-cell development, gamma secretase inhibitors (GSI), targeted therapies

## Abstract

T-cell acute lymphoblastic leukemia (T-ALL) is an aggressive hematological disease that arises from the oncogenic transformation of developing T cells during T-lymphopoiesis. Although T-ALL prognosis has improved markedly in recent years, relapsing and refractory patients with dismal outcomes still represent a major clinical issue. Consequently, understanding the pathological mechanisms that lead to the appearance of this malignancy and developing novel and more effective targeted therapies is an urgent need. Since the discovery in 2004 that a major proportion of T-ALL patients carry activating mutations that turn *NOTCH1* into an oncogene, great efforts have been made to decipher the mechanisms underlying constitutive NOTCH1 activation, with the aim of understanding how NOTCH1 dysregulation converts the physiological NOTCH1-dependent T-cell developmental program into a pathological T-cell transformation process. Several molecular players have so far been shown to cooperate with NOTCH1 in this oncogenic process, and different therapeutic strategies have been developed to specifically target NOTCH1-dependent T-ALLs. Here, we comprehensively analyze the molecular bases of the cross-talk between NOTCH1 and cooperating partners critically involved in the generation and/or maintenance and progression of T-ALL and discuss novel opportunities and therapeutic approaches that current knowledge may open for future treatment of T-ALL patients.

## 1. NOTCH Signaling Pathway

NOTCH comprises a family of highly evolutionary conserved membrane-bound receptors (NOTCH1-4 in mammals), which regulate essential functions in a wide variety of cell types within the organism, such as proliferation, differentiation, or apoptosis, thus controlling fundamental processes in embryonic and adult life, including tissue homeostasis, cell fate decisions or tissue development and regeneration [1]. NOTCH signaling is initiated by cell-to-cell contacts that allow the interaction between NOTCH receptors expressed on the surface of one cell and their ligands, belonging to the DELTA-LIKE (DLL1, DLL3 and DLL4) and JAGGED families (JAG1 and JAG2) in vertebrates [2], expressed on neighboring cells (Figure 1). Ligand recognition by NOTCH receptors triggers a proteolytic cleavage cascade leading to irreversible activation of the pathway. Firstly, ectodomain shedding, mediated by metalloproteases of the ADAM family, occurs within the heterodimerization domain (HD) at the extracellular portion of NOTCH (site-2); and secondly, a protein complex with intrinsic γ-secretase activity enzymatically cleaves the transmembrane domain (TM; site-3), and releases NOTCH intracellular domain (NICD). NICD then translocates to the nucleus and binds to the CSL (CBF1/RBPJ, Su(H), Lag-1) transcription factor, turning it from a repressive into a transcriptionally active complex by displacement of corepressors [SHARP, histone deacetylase-1 (HDAC-1), silencing mediator of retinoic acid and thyroid hormone receptors (SMRT), N-CoR], and recruitment of coactivators [Mastermind-like 1-3 (MAML1-3)] and histone acetyltransferases (p300/CBP) [2]. Several cellular- and context-dependent NOTCH transcriptional targets are then induced, including *HES1-5*, *HEY-1* and *-2*, *PTCRA*, *MYC*, *DTX1*, *IL7R*, *CD44* and *RUNX1* [3]. The shutdown of NICD signaling is finally induced following the recruitment of cyclin-dependent kinase 8 (CDK8) by MAML1 and phosphorylation of residues within the PEST domain [4]. Phosphorylated NICD can then be recognized by FBXW7, an E3 ubiquitin ligase that poly-ubiquitinates NICD and targets it for proteasomal degradation. Several other E3 ubiquitin ligases modulate NOTCH activity in different cellular contexts, either by ubiquitination of NOTCH receptors or ligands, such as suppressor of Deltex [Su(dX)], NEDD4, and MIB1/2 [5]. Recently, deubiquitinases, such as ubiquitin-specific peptidase 7 (USP7) and 10 (USP10), were shown to biochemically interact with and deubiquitinate and stabilize the intracellular NOTCH1 domain, contributing to sustain pathway activation. Thus, inhibition of endogenous USP7 activity results in decreased NOTCH1 protein levels [6,7], and USP10 inactivation also reduces NICD stability and diminishes NOTCH1-induced target gene expression [8], through a mechanism dependent on the hydroxylation status of NOTCH1 [9].

## 2. Oncogenic Mutations and Deregulation of NOTCH1 Signaling in T-cell Acute Lymphoblastic Leukemia

T-cell development is one of the best-studied functions of NOTCH signaling in vertebrates. NOTCH ligands expressed by thymic epithelial cells (TECs) provide unique signals required at successive maturation stages for the commitment and development of bone marrow- (BM-) or fetal liver-derived HPCs along the T-cell lineage [10,11,12,13,14,15,16]. Seminal work by Radtke and colleagues demonstrated that the T-cell fate specification of multipotent HPCs is strictly dependent on signaling mediated by the NOTCH1 receptor [17], as conditional *Notch1* loss-of-function mice had a defective T-cell production, and accumulated non-T cell lineages including B [17] and dendritic cells [18] in the thymus. Conversely, constitutive expression of active NOTCH1 in HPCs impaired B-cell differentiation and promoted the appearance of ectopic double positive (DP) thymocytes at extrathymic locations, such as BM and peripheral blood (PB) [19,20]. Once T-lineage commitment is induced, maintenance of T-cell specification and further differentiation is dependent on recurrent NOTCH1 signaling [11]. Separate studies showed that DLL4 is the specific NOTCH1 ligand indispensable for triggering both T-cell commitment and differentiation [13,21], although other NOTCH ligands (and receptors) are expressed in the thymus and participate in other processes away from T-cell development [16,22,23,24,25,26].

NOTCH1-DLL4 interactions control the massive expansion that T-cell progenitors undergo at two critical intrathymic checkpoints. At the early CD4^−^ CD8^−^ double negative (DN1-3) stages, thymocyte survival and proliferation are greatly dependent on interleukin 7 (IL-7) produced by TECs. Later, when functional TCRβ rearrangements occur, a TCRβ chain assembles together with the pTα invariant chain and CD3 modules at the cell surface to form the pre-T-cell receptor (pre-TCR), which then signals for survival and proliferation during β-selection [27,28]. NOTCH1 signaling critically contributes to both checkpoints by promoting the transcription and expression first of the *IL7R* gene encoding the α chain of the IL-7 receptor (IL-7Rα) [29,30,31], and then of the *PTCRA* gene encoding the pTα chain of pre-TCR [32]. Later in development, NOTCH1 signaling has to be downregulated at the CD4^+^ CD8^+^ double positive (DP) stage to allow for αβ T-cell development [15], and NOTCH1 further regulates the CD4 versus CD8 single positive (SP) lineage decision by modulating TCR signaling in DP thymocytes [33,34]. These results highlight the essential role of the NOTCH1 pathway as a master regulator of thymocyte survival and proliferation at crucial checkpoints. It was thus predictable that deregulated NOTCH1 signaling could lead to the aberrant expansion of developing thymocytes and the generation of leukemia. However, the first identified genetic alteration involving the *NOTCH1* gene was detected only in less than 1% of T-cell acute lymphoblastic leukemia (T-ALL) patients. It consists of a chromosomal translocation to the *TRB* locus [t(7;9)(q34;q34.3)], which generates a truncated and constitutively active form of NOTCH1 that lacks site-2 and site-3 cleavage sites [35,36], resulting in γ-secretase-independent activation. Therefore, the seminal discovery in 2004 that more than 60% of T-ALLs carry *NOTCH1* gain-of-function (GOF) somatic mutations that also generate a truncated and constitutively active receptor caused an immense revolution in the field, pointing to NOTCH1 as a key regulator of T-ALL pathogenesis [37].

T-ALL is a heterogeneous tumor originating from the malignant transformation of developing T cells that arises from accumulating genetic alterations in key oncogenic, tumor suppressor and developmental pathways, which confer uncontrolled cell proliferation and survival capacity. Notably, NOTCH1 mutations producing aberrant NOTCH1 activation are present in T-ALLs from all major molecular subtypes [38,39]. The classification of clinically relevant biological groups of T-ALL was originally based on the immunophenotypic resemblance of the tumor to particular T-cell maturational stages, indicative of a specific developmental arrest during T-cell transformation. Currently, however, accuracy in the classification and stratification of T-ALL patients preferentially relies on genetic and transcriptomic studies of the specific mutational profile, gene expression signatures and genomic alterations of T-ALL samples [40,41,42,43,44,45,46]. Early T-lineage progenitor (ETP) T-ALL, which is characterized by a very immature CD4^−^ CD8^−^ DN T-cell phenotype and the coexpression of stem cell and myeloid-associated markers [47], commonly harbor mutations in epigenetic regulators (*EZH2*, *IDH1*, *IDH2*, *DNMT3A*), transcription factors associated to hematopoietic and T-cell development (*ETV6*, *GATA3*, *RUNX1*), and signaling pathways such *NRAS* and *FLT3* [48]. Cortical T-ALLs, characterized by the expression of CD1a and a DP phenotype, are typically associated with genetic aberrations in *TLX1*, *TLX3*, *NKX.2.1* and *NKX.2.2* homeobox genes, and in *CDKN2A*, *TAL1* and *LMO1,2* [40,49]. *TAL1* and *LMO1,2* expression are also typical of mature T-ALL expressing surface CD3 [40,50]. Interestingly, *NOTCH1* oncogenic mutations are frequent in cortical T-ALLs and can be found in up to 40% of mature T-ALLs [50], while ETP T-ALLs have a lower prevalence of *NOTCH1* mutations [42].

T-ALL activating NOTCH1 mutations are mostly located in the HD or in the LIN12-Notch repeats (LNR) domain within the negative regulatory region (NRR) of the receptor located at the extracellular region. Class I HD mutants disrupt the HD domain, while class II involve short insertions that displace and expose the ADAM-dependent cleavage site-2, allowing for constitutive ligand-independent activation of NOTCH1. Sets of insertions that expand the juxtamembrane region (JME mutants) have also been described [51]. Additionally, a different type of mutation compromises the FBXW7 ubiquitin ligase-binding site located in the C-terminal PEST domain of NICD [37], which impairs the proteasomal degradation of NICD while increasing the half-life. Concordantly, FBXW7 mutants unable to bind NICD have been described to maintain NOTCH1 signaling in T-ALL and to confer resistance to γ-secretase inhibitors (GSI) [52,53]. In addition, high expression of USP7 has been shown to correlate with NOCH1 stabilization and expression in NOTCH1-dependent T-ALLs [6,7]. Although all activating *NOTCH1* mutations confer constitutive and/or prolonged NOTCH1 signaling, only some of them, such as those leading to constitutive expression of the NICD form of NOTCH1, promote “strong” NOTCH1 signaling and have the intrinsic capacity to induce T-ALL in murine models [19]. Others, including HD and PEST mutations found in patients, generate weaker NOTCH1 mutant forms that complement other leukemogenic signals or require cooperation with additional oncogenic pathways or with loss-of-function tumor suppressor mutants for driving T-cell transformation [54,55,56,57,58]. These findings support the view that *NOTCH1* mutations are secondary oncogenic events in many T-ALL patients [59]. However, *NOTCH1* mutations can also act as the initial genetic event driving T-ALL [60]. More importantly, even when NOTCH1 participates secondarily, NOTCH1 activation is an early hallmark of T-cell leukemogenesis and a key regulator of the leukemia-initiating cell (LIC) activity of T-ALL [61,62]. Accordingly, T-ALL tumors show “addiction” to NOTCH1 signaling, reflecting a high selective pressure for NOTCH1 activation [63].

### NOTCH1 Targeting Therapeutic Strategies

The realization that NOTCH1 signaling is a major oncogenic driver of T-ALL generated much expectation about the efficacy of pan-NOTCH inhibitors for the treatment of this disease. Several interventional therapies based on compounds that target the activation of NOTCH signaling at different levels have been developed and tested in vitro and in preclinical models (Figure 1, Table 1) [64]. Initial studies demonstrated the therapeutic efficacy of GSIs. However, the observation of severe adverse effects, particularly intestinal toxicities, in the first patients orally receiving GSIs [65,66] hampered the implementation of this therapy. Although these adverse effects could be minimized by coadministration of glucocorticoids [67], and novel GSI compounds have recently entered into clinical trials (i.e., AL101, AL102, PF-03084014, and RO4929097 [64,68,69,70]; Figure 1, Table 1), GSI treatment is still not widely established in the clinic. Indeed, PF-03084014 (Nirogacestat) was evaluated in a Phase I trial, including relapsed/refractory T-ALL patients, and although well tolerated, no conclusive anti-leukemic effects were obtained [70]. AL101, a potent GSI showing very promising anti-tumor efficacy in preclinical models [69]), is currently being evaluated against triple-negative breast cancer (NCT04461600) and adenoid cystic cancer (NCT04973683, NCT03691207), but no T-ALL trials have been opened. On the other hand, RO4929097 was well tolerated in several trials against different types of cancer, but their therapeutic efficacy in T-ALL patients has not been established. Moreover, NOTCH1-mutant but GSI-resistant T-ALLs demand the implementation of alternative strategies that efficiently and safely target NOTCH1-dependent tumor cells without detrimental effects. Promising monoclonal antibodies have been described to specifically bind mutated NOTCH1 receptors [i.e., OMP-52M51 (Brontictuzumab) [71], MAb604.107 [72]], or to block NOTCH1-ligand interactions (i.e., anti-DLL4, Demcizumab [73] and Enoticumab [74]) but their clinical relevance is yet to be established. The safety and tolerability of anti-NOTCH1 OMP-52M51 antibody have been evaluated in clinical trials for refractory solid tumors (NCT01778439) [75] and relapsed/refractory lymphoid malignancies, including T-ALL (NCT01703572) [76]. Both studies have found that OMP-52M51 has moderate anti-tumor activity and is well tolerated, with diarrhea as the main primary toxicity. Regarding anti-DLL4 antibodies, Demcizumab (NCT01189968) [77] and Enoticumab (NCT00871559) [78] have been assayed in Phase Ia trials for advanced solid tumors, showing partial responses and disease stabilization.

Other molecules shown to inhibit NOTCH activation in preclinical settings include ADAM inhibitors [79] and NOTCH decoys [80]. MAML1 and NOTCH ternary complex (NTC) inhibitors, such as SAHM1 [81] and NADI-351 [82], respectively, capable of blocking NOTCH transcriptional complex activity, have demonstrated potential anti-cancer properties. Inducing NOTCH1 intracellular domain degradation by inhibition of endogenous USP7 activity, either by shRNA or with pharmacological inhibitors (i.e., P2207, P217564) [6,7], has been shown to reduce in vitro proliferation, to induce cytotoxicity in T-ALLs, and to inhibit in vivo expansion of NOTCH1-dependent T-ALLs. Novel strategies directed to impair NOTCH1 receptor maturation processes in the endoplasmic reticulum (ER)/Golgi compartments, and subsequent cell surface expression, have proved very efficient in blocking NOTCH1 activation. Among them, SERCA (sarco/endoplasmic reticulum calcium ATPase) inhibitors have been identified as potential regulators of leukemia-associated NOTCH1 mutant signaling and cell cycle progression and expansion of T-ALL xenografts [83]. Improved formulations of SERCA inhibitors, such as CAD204520, have been developed aimed at reducing off-target toxicity or at inducing specific delivery to leukemic cells [84]. Other inhibitors that interfere with the maturation of NOTCH receptors, such as fucose analogs (6-alkynyl and 6-alkenyl fucose), which interfere with the *O*-fucosyltransferase 1 (POFUT1)-dependent transfer of *O*-fucose to epidermal growth factor (EGF) repeats of NOTCH receptors, have proved to successfully impair DLL1 and DLL4 interactions with NOTCH receptors [85]. On the other hand, recent phosphoproteomic and expression-based screenings have revealed novel targetable kinases potentially involved in the NOTCH1-launched oncogenesis program and in GSI resistance [86,87,88,89,90,91]. Some examples are Src-family kinases and active cyclin-dependent kinases [90] and protein kinase C (PKC) delta [89].

Besides therapeutic strategies directly targeting single components of the NOTCH1 pathway itself (Table 1), the realization that the T-ALL oncogenic process involves multiple NOTCH1 collaborators has revealed new molecular vulnerabilities that offer the opportunity for novel therapeutic developments. NOTCH1 oncogenic collaborators include cell cycle regulators, proliferation and adhesion receptors, chemokine receptors, metabolic regulators or RNA modulators (see below). Therefore, understanding the molecular programs underlying NOTCH1 signaling-dependent T-cell transformation and its multiple steps of regulation is critical for designing new drug combinations that target aberrant NOTCH1 activation in T-ALLs. In addition, deciphering the mechanistic bases of NOTCH1 cross-talk with other signaling pathways and molecular regulators involved in cooperative oncogenesis will help to delineate novel therapeutic approaches to specifically target T-ALL cells while diminishing the risk of undesired collateral effects on healthy cells.

## 3. Molecular Collaborators of NOTCH1 Signaling in T-ALL Pathogenesis

Multiple signaling pathways and molecular regulators collaborate with NOTCH1 signaling, contributing to T-ALL pathogenesis through several mechanisms, such as favoring cell proliferation and survival through specific surface receptors or by impaired cell cycle control, increasing the fitness of leukemic cells by the modulation of their metabolic demands, or guiding T-ALL cells to specialized niches where they can expand uncontrollably. In this section, we will overview recent findings uncovering some of the molecular collaborators that cooperate in NOTCH1-mediated T-ALL pathogenesis (Figure 2, Table 2). Most experimental strategies approaching this aim have relied on murine models of NOTCH1-mediated T-cell oncogenesis. Widely used is the mouse T-ALL model established by Pear and colleagues, consisting in the transplantation to host mice of murine BM HPCs transduced with the constitutive active form of NOTCH1 (NICD) [19]. The model allows for the generation and follow-up of a murine T-ALL, opening the door to preclinical testing of new therapeutic options to fight disease relapse, which is a main T-ALL clinical issue. However, whether the model recapitulates the earliest stages of human T-ALL pathogenesis has remained elusive, as a retrospective analysis of human T-ALL onset is unfeasible in patients, and no information on such early stages is yet available. Overcoming this limitation, a novel model of human T-ALL generation has recently been developed, consisting of the reconstitution of NSG (NOD.Cg-*Prkdc^scid^ Il2rg^tm1Wjl^*/SzJ) immunodeficient mice with human CD34^+^ umbilical cord blood (CB) hematopoietic progenitors expressing constitutively active NOTCH1 [92]. Transplanted mice generate *de novo* clonal human leukemia that resembles T-ALL in patients, allowing for the delineation of the pathogenic events associated with the onset of the human disease, and revealing the stepwise impact of NOTCH1 mutations on human T-ALL pathogenesis. An additional human T-ALL model has also been described by Weng’s group in NSG mice transplanted with CD34^+^ CB HPCs cotransduced with NOTCH1-ΔE mutants together with a combination of oncogenes identified in T-ALL patients, i.e., *LMO2*, *TAL1* and *BMI1* (LTB) [93], allowing to study the collaboration of these mutations with the NOTCH1 oncogenic program. Therefore, mouse and human T-ALL models provide clinically relevant complementary tools useful for the characterization of NOTCH1 functional interactions with other signaling pathways and molecular regulators in T-ALL, for the identification of human T-ALL LICs and their specific targets, and for the development and validation of improved therapeutic options.

### 3.1. RNA Regulators

In recent years, the non-coding component of the genome has gained great relevance because of its implication in numerous physiological and pathological processes. Non-coding RNAs (ncRNAs) include microRNAs (miRNAs), long non-coding RNAs (lncRNAs), circular RNAs (circRNA), PIWI-interacting RNAs (piRNAs), small nucleolar RNAs (snoRNAs), transcribed ultraconserved regions (T-UCRs), and large intergenic non-coding RNAs (lincRNAs) [94]. While miRNAs have been the most studied ncRNAs in T-ALL to date, lncRNAs and circRNAs have drawn attention lately because of their contribution to T-ALL pathogenesis, and especially to NOTCH1-dependent T-ALL. However, the relationship between NOTCH signaling and ncRNA deregulation in T-ALL is still a matter of intensive study.

**Table 2 ijms-24-01383-t002:** RNA regulators and signaling pathways contributing to NOTCH1-induced T-ALL pathogenesis.

Cellular Process	Molecule	Experimental Model	Effects on T-ALL	References
**RNA regulators**	miR-223	Cotransduction with NICD in FL-HPCs	Accelerated NICD-dependent leukemia onset	[95]
	miR-19	Cotransduction with NICD in FL-HPCs	Accelerated NICD-dependent leukemia onset	[96]
	miR-181a/b	*miR-181ab1-/-* HPCs + NICD	Increased mice survival & Reduced leukemic burden	[97]
	miR-30a	Retroviral overexpression in T-ALLs	Decreased T-ALL growth rate in vitro	[98]
	lncRNAs (*LUNAR1*)	*LUNAR1* shRNA + T-ALL cell lines	Growth disadvantage	[99]
	circRNAs	circ_0000745 overexpression	Promoted proliferation Reduced apoptosis	[100]
**Survival/** **Proliferation**	IGFR1	*IGF1R^neo^* BM HPCs + NICD	Reduced LIC activity	[101]
	IL-7R	*Il-7r-/-* BM HPCs + NICD	In vivo T-ALL impairment	[102]
	pre-TCR	*Rag2-/-, Slp76-/-* BM HPCs + NICD	In vivo T-ALL impairment/delay	[103,104]
**Metabolic regulators**	KRAS	*Kras^G12D^* BM HPCs *+* NOTCH1 GOF alleles	Accelerated T-ALL development	[54]
	PI3K/AKT/ mTOR	*Pten*-loss-of-function + L1601P Δ-PEST NOTCH1	Acquisition of in vivo GSI resistance	[105]
		GSI + mTOR inhibitor	Decreased in vitro T-ALL proliferation Increased mice survival	[106]
				[107]
	MYC	*N-Me-/-* BM HPCs + ΔE-NOTCH1	In vivo T-ALL impairment	[108]
**Cell cycle** **regulators**	CYCLIN D3	*Ccnd3-/-* HPCs + NICD	In vivo T-ALL impairment	[109]
	CDK6	*Cdk6-/-* HPCs + NICD	In vivo T-ALL impairment	[110]
**Chemotaxis/Adhesion**	CD44	Anti-CD44 antibody	Impaired BM engraftment and T-ALL progression	[92]
	CXCL12	Vascular endothelial cell-specific *Cxcl12-/-* mice	Reduced Notch-dependent T-ALL expansion in vivo	[111]
	CXCR4	AMD3100 inhibitor *Cxcr4-/-* FL-HPCs + NICD	Blocked T-ALL BM colonization In vivo T-ALL impairment	[112]
	CCR7	*Ccr7-/-* BM HPCs + NICD	Inhibition of CNS infiltration	[113]
	CCR5/9	GSI/siRNA NOTCH1	Reduced T-ALL proliferation/migration	[114]

Abbreviations: T-ALL (T-cell acute lymphoblastic leukemia); NICD (Notch intracellular domain); FL (fetal liver); HPCs hematopoietic progenitor cells); shRNA (short hairpin RNA); BM (bone marrow); GSI (γ-secretase inhibitor); mTOR (mammalian target of rapamycin); GOF (gain of function); siRNA (small interference RNA); CNS (central nervous system).

### 3.2. miRNAs

The significance of the cross-talk between NOTCH1 and miRNAs in T-ALL was revealed by Junker and colleagues [115], who took advantage of a T-cell-specific *Dicer1* conditional loss-of-function mouse model to demonstrate the strict requirement of the miRNA machinery for the generation of NOTCH1-dependent T-ALL. DICER1 is an RNaseIII endoribonuclease that cleaves a precursor miRNA into the 20-22 nucleotide (nt)-long mature regulatory miRNA that will then be incorporated into the RNA-induced silencing complex (RISC), together with the targeted messenger RNA (mRNA), to prevent its translation or to induce its degradation [116]. Loss of miRNAs induced apoptosis in leukemic T cells and impaired leukemia development and progression both at early- and late-stages of T-ALL pathogenesis [115]. Accordingly, several miRNAs (miR-19b, miR-20a, miR-26a, miR-92, and miR-223) accelerate leukemia progression when coexpressed with NICD in a mouse T-ALL model [95] through a mechanism involving specific targeting of T-ALL tumor suppressor genes including *Pten*, *Bim*, *Nf1*, *Fbxw7*, *Ikzf1* and *Phf6.* Hence, these miRNAs were categorized as T-ALL oncomiRs. Supporting this oncogenic role, miR-19 expression was 5- to 17-fold-increased in T-ALL samples when compared to normal T lymphocytes [96]. In addition, genetic analyses showed that miR-19 is involved in two T-ALL-associated gene rearrangements, comprising T-cell receptor α/γ (*TRA/D*) locus (t13;14)(q32;q11) and *NOTCH1* locus t(9;14)(q34;q11), the latter generating a constitutively active form of NOTCH1. When expressed together with NICD in transplanted fetal liver (FL) HPCs, miR-19 significantly decreased mouse mean survival compared to mice transplanted with HPCs overexpressing NICD alone, suggesting that miR-19 exacerbates the NOTCH1-induced oncogenic program [93]. The identification of several miR-19 targets, including *Bim*, *Pten*, *Prkaa1*, and *Ppp2r5e*, suggests that miR-19 may sustain lymphocyte survival contributing to leukemogenesis through the regulation of phosphoinositide-3 kinase (PI3K)-related pathways [96].

Several reports have included miR-223 in the list of regulators that cooperate with oncogenic NOTCH1 in T-ALL. Besides its contribution to myeloid development, particularly to granulocyte and macrophage differentiation and to the pathogenesis of acute myeloid leukemia (AML) [117], miR-223 is induced by active NOTCH1 and participates in T-ALL leukemogenesis by repressing the tumor suppressor FBXW7 [118]. Likewise, miR-223-dependent FBXW7 suppression has been described in T-ALLs induced by the oncogenic transcription factor TAL-1 [119]. Accordingly, an inverse correlation of miR-223 and FBXW7 expression has been observed in a panel of T-ALL patient-derived xenografts (PDXs) [118]. In addition to FBXW7, miR-223 targets other regulators of NOTCH1 degradation, such as ARRB1 (Arrestin β1), which serves as an E3 ubiquitin ligase adaptor and facilitates NOTCH1 ubiquitination and degradation via interaction with DTX1 [120]. However, multiple levels of cross-regulation between miR-223 and NOTCH1 may contribute to T-ALL, considering that NOTCH1 signaling can also repress miR-223 leading to upregulation of insulin-like growth factor-1 receptor (IGF1R), which is important for LIC activity in T-ALL [101,121].

Other miRNAs collaborating in NOTCH1-dependent T-ALL pathogenesis include miR-181a-1/b-1 and miR-30a. Deletion of miR-181a-1/b-1 impairs T-cell leukemogenesis induced by either murine NICD or human P12ΔP NOTCH1 alleles, reducing leukemic burden within the thymus and increasing up to 80% mouse survival [97]. At the molecular level, miR-181a-1/b-1 controls the strength and threshold of NOTCH1 receptor activity by dampening multiple negative feedback regulators downstream of both NOTCH1 and pre-TCR [97]. In contrast to the aforementioned miRNAs, miR-30a exhibits tumor-suppressive properties in vitro and inhibits NOTCH1 and NOTCH2 expression in an MYC-dependent manner, and it was also found to participate in a bidirectional regulatory circuitry between NOTCH and MYC in T-ALL [98]. Collectively, available information points to several miRNAs as optimal therapeutic targets for the treatment of NOTCH1-dependent T-ALLs (Figure 3).

### 3.3. lncRNAs

lncRNAs are a heterogeneous class of ncRNAs that, basically, have a 200 nt minimum length and an apparent lack of protein-coding potential [122]. Although their mechanisms of action are not fully understood, lncRNAs are known to participate in several cellular processes, such as cell cycle progression [123] and epigenetic regulation [124] in different contexts, including hematopoiesis [125,126] and leukemia [127,128,129,130]. Several reports have addressed the NOTCH1-modulated lncRNA repertoire of normal thymocytes and T-ALLs [99,131]. By comprehensive mapping of lncRNAs, Trimarchi and colleagues identified a T-ALL-specific lncRNA signature, including more than 1000 transcripts [99]. Chromatin immunoprecipitation sequencing (ChIP-seq) and GSI treatment assays revealed that a great proportion of these transcripts were regulated by NOTCH1. Of note, expression of the lncRNA, *LUNAR1*, which displays T-ALL anti-tumor potential, was regulated by NOTCH1 through the *IGF1R* locus enhancer [99]. In a different study, NALT lncRNA was found to correlate with NOTCH1 expression in pediatric T-ALL and to regulate the NOTCH1 signaling pathway through *cis*-regulatory mechanisms [132]. Since lncRNAs still remain widely unexplored, it is expected that a large number of new molecules will soon be identified as part of the gene expression network that oncogenic NOTCH1 shapes in T-ALL pathogenesis.

### 3.4. circRNAs

circRNAs comprise a novel group of ncRNAs generated from back-splicing when an exon is spliced to the preceding instead of to the downstream exon, resulting in a covalently-closed continuous loop structure [133]. Most of them are expressed at low levels in normal cells; however, the higher expression of some circRNAs, and their regulation during cell differentiation processes, suggest a specific yet unknown function. Very few circRNAs have been shown to date to contribute to NOTCH1-dependent oncogenesis. The T-ALL circRNA landscape has been established recently [134], identifying almost 1000 circRNAs with differential expression in T-ALL associated with particular T-ALL genetic signatures. However, their relationship with the NOTCH1 pathway remains to be elucidated. Only circ_0000745 was shown to favor NOTCH1 signaling in T-ALL, acting as a sponge for miR-193-3p and releasing miR-193-3p-mediated NOTCH1 repression through a competing endogenous RNA (ceRNA) mechanism [100].

### 3.5. Survival/Proliferation Signaling Receptors

#### 3.5.1. Interleukin 7 Receptor (IL-7R)

Interleukin 7 (IL-7) is the essential thymic epithelium-derived cytokine sustaining the first wave of extensive proliferation that immature T cells must undergo during their development in the thymus. Mice deficient in IL-7 (*Il7-/*-) or in its receptor (*Il7r-/-*) have an early block in T-cell development, with a dramatic reduction in the generation of mature T cells, and also show an impaired B lymphopoiesis [135,136,137]. In humans, IL-7 receptor (IL-7R) paucity results in a complete lack of T cells, leading to severe combined immunodeficiency (SCID), but does not affect B-cell development [138,139]. IL-7R is commonly expressed in T-ALL, reflecting a developmental block of T-cell lymphoblasts at an IL-7-dependent developmental stage. Accordingly, IL-7-dependency is shared by most T-ALL cells expressing functional IL-7Rs, which respond to IL-7 stimulation by activating Janus kinase (JAK)/signal transducer and activator of transcription (STAT), mitogen-activated protein kinase (MAPK)/extracellular signal-regulated kinase (ERK) and PI3K/protein kinase B (PKB or AKT)/mammalian target of rapamycin (mTOR) signaling pathways that lead to cell survival and proliferation. Supporting a prominent role for IL-7R in physiological and pathological NOTCH1-dependent T-cell development, NOTCH1 signaling was shown to directly control IL-7R-dependent T-cell expansion by transcriptionally regulating the expression of the IL-7R alpha (IL-7Rα) subunit, via binding to a promoter [29] and enhancer [30,31] regions in the *IL7R* gene. Also, NOTCH1-mediated T-ALL cell proliferation was shown to rely, to a great extent, on IL-7R signals, as the defective expansion of GSI-treated T-ALL cell lines could be rescued by enforced activation of IL-7/IL-7R signaling [29]. Importantly, it has been recently shown that IL-7R expression is an early functional biomarker of T-ALL cells with LIC potential [102]. Inhibition of endogenous IL-7Rα expression by short-interfering RNA (shRNA) was shown to impair engraftment and progression of IL-7R^+^ primary human T-ALLs [102], highlighting the relevance that targeting physiological IL-7R signaling may have in future T-ALL therapeutic interventions. More importantly, NOTCH1-dependent T-ALL pathogenesis seems strictly dependent on the IL-7/IL-7R axis, given the failure of NICD to induce T-cell transformation from *Il7r-/-* BM HPCs [102].

Besides its cooperation with NOTCH1, *IL7R* can behave itself as an oncogene in T-ALL. Gain-of-function mutations in *IL7R* exon 6, consisting of in-frame insertions or deletions-insertions, have been described in approximately 10% of T-ALL patients from different cohorts [140,141]. These mutations introduce an unpaired cysteine in the extracellular juxtamembrane-transmembrane region, promoting de novo formation of intermolecular disulfide bonds between mutant IL-7Rα subunits, which drive constitutive IL-7-independent proliferative and survival signaling and contribute to cell transformation and aberrant T-ALL growth. Other studies have shown that wild-type IL-7R overexpression is also able to induce leukemia with features of human T-ALL in mice. Either tetracycline-inducible *Il7r* transgenic mice or T-cell-specific human *IL7R* knockin mice (*Rosa26-hIL-7R.huCD2-Cre*) showed aberrant T-cell differentiation that resulted in the expansion of immature T cells in peripheral organs as mice aged and eventually led to the development of transplantable T-cell malignancies [142]. These data support a prominent pro-oncogenic role of the IL-7R pathway in T-ALL, which concurs with the identification of oncogenic mutations in several IL-7R-signaling downstream molecules such as *JAK1*, *JAK3*, and *STAT5B* in 5–20% of T-ALL cases [43,48,143]. Thus, either by itself or in combination with NOTCH1 signaling, IL-7R represents a major pathogenic pathway in T-ALL and a promising therapeutic target. Accordingly, recently developed anti-IL-7R antibodies capable of binding both wild-type and mutant IL-7Rs have shown therapeutic efficacy in preclinical models, not only by delaying IL-7R^+^ T-ALL development and reducing tumor burden but also by exerting therapeutic action against chemotherapy-resistant relapsing disease [144,145]. Additionally, JAK inhibitors have shown promising anti-leukemic effects in different assays [86,142] and are currently being tested in clinical trials in combination with other chemotherapeutic drugs (Figure 3).

#### 3.5.2. Insulin Growth Factor 1 Receptor (IGF1R)

Insulin growth factors 1/2 (IGF1/2) bind to the transmembrane receptor IGF1R and activate downstream signaling pathways, including PI3K/AKT/mTOR and RAS/MEK/ERK, regulating proliferation and survival of normal [146] and leukemic stem cells [147]. The contribution of IGF1R to NOTCH1-dependent T-ALL pathogenesis was established using an *IGF1R* conditional mouse model with reduced expression of IGF1R (*IGF1R^neo^*) [101]. BM cells from *IGF1R^neo/neo^* mice transduced with NOTCH1-ΔE were able to develop leukemia, but their LIC activity was dramatically reduced compared to that of wt transduced BM cells, and T-ALL arising in first recipients was barely transplantable in secondary and tertiary hosts. Molecular analysis identified an NICD/CSL-binding site within intron 20 of *IGF1R*, suggestive of the presence of a NOTCH1-responsive enhancer, through which NOTCH1 contributes to maintaining high levels of IGF1R expression, rendering T-ALLs more efficient in responding to low levels of IGF1/2 [101]. Additionally, as described above (Section 3.1), both miRNAs (miR-223) [121] and lncRNAs (*LUNAR1*) [99] can mediate the regulation of NOTCH1 over IGF1R, further supporting that this growth-factor receptor pathway plays an important role in NOTCH1-dependent leukemogenesis.

#### 3.5.3. Pre-TCR

During T-cell development, thymocytes that undergo a productive rearrangement at the T-cell receptor β (*TRB*) locus express a TCRβ chain that couples to the invariant pTα chain and CD3 molecules (CD3δ, ε, γ and ξ) in the endoplasmic reticulum to form a pre-TCR complex that is exported to the cell surface [148,149]. Early activation events downstream of pre-TCR signaling include the phosphorylation of p56^LCK^, zeta-chain-associated protein kinase 70 (ZAP-70) and SH2 domain-containing leukocyte protein of 76 KDa (SLP-76), which in turn activates PI3K/AKT, phospholipase C (PLC)γ/nuclear factor of activated T cells (NFAT), PKC and RAS/MAPK pathways [150]. Proper pre-TCR signaling is mandatory to induce the survival and proliferation of committed pre-T cells and their differentiation towards the DP TCRαβ stage, a process known as β-selection [27,28]. Accordingly, T-cell maturation is hindered in mice deficient in any of the components of the pre-TCR (i.e., pTα [151], TCRβ [152], or CD3 chains [153]) or in mice with impaired pre-TCR signaling [154,155]. For many years, pre-TCR has been considered a constitutively active complex, which signals in the absence of ligand by means of oligomerization at the cell membrane [156] and undergoes constitutive internalization and lysosomal degradation for signaling termination [157,158]. In recent years, however, seminal studies by Reinherz’s group have challenged this view and showed that the pre-TCR binds self-peptide-major histocompatibility complex (MHC) complexes at the β-selection checkpoint [159,160,161] and conforms immunological synapses dependent on NOTCH1 and C-X-C chemokine receptor type 4 (CXCR4) signaling [162].

NOTCH1 and pre-TCR act in close collaboration at several levels during T-cell development. Firstly, the *Ptcra* gene, which encodes the pre-TCR pTα chain, is a direct NOTCH1 transcriptional target [32]. Additionally, mice with a T-cell-specific inhibition of NOTCH activity have a reduction in pre-TCR-associated events and a block in the DN/DP transition, which cannot be rescued by *Tcrb* or *Tcra/b* transgene overexpression [163], suggesting that NOTCH1 and pre-TCR act in parallel during β-selection. This cooperation has also been demonstrated in vitro [164]. The mechanism underlying this cross-talk is not completely understood, but NOTCH1 is known to activate a PI3K/AKT-dependent metabolic program to regulate pre-TCR-induced cell growth [165]. NOTCH1 also regulates the expression of FBXL1, and pre-TCR induces FBXL12, two factors of the SCF E3 ligase complex, which promote cyclin-dependent kinase 1B (CDKN1B) degradation resulting in cell cycle progression and proliferation of β-selected thymocytes [166]. While NOTCH1 activation is necessary at the onset of pre-TCR signaling, it needs to be downregulated after β-selection, later at the DN to DP transition, to avoid aberrant DP thymocyte proliferation that could lead to leukemogenesis [103,167,168]. To extinguish NOTCH1 signaling, pre-TCR induces the expression of *Bcl6*, a transcriptional repressor that diminishes NOTCH1-mediated transcription and NOTCH1 activation [169] and may have a tumor suppressor role in NOTCH1-mediated T-ALL pathogenesis [170]. Pre-TCR signaling also inhibits *NOTCH1* transcription via up-regulation of the E2A inhibitor ID3 [171]. These data reveal a direct link between pre-TCR signaling and NOTCH1 expression and suggest that molecular strategies exist during thymocyte development for inhibiting NOTCH1 signaling in pathologic conditions. This is particularly relevant given that NOTCH-induced leukemogenesis in mice has been shown to be dependent on a functional pre-TCR. Indeed, NICD-transduced BM HPCs from mice with impaired pre-TCR function (i.e., *Rag2-/-, Slp76-/-*) do not develop leukemia [103] or show a delayed leukemic onset [104] depending on the retroviral construct used, and the aggressiveness of NOTCH3-mediated oncogenesis is pre-TCR-dose-dependent as well [172]. In this regard, *PTCRA* transcription and protein expression have been detected in a high proportion of human T-ALL cases and T-ALL cell lines at different maturational stages [172,173,174]. Also, by comparative immunophenotype, it has been proposed that about half of the αβ-lineage primary human T-ALLs might express the pre-TCR at the cell surface. The availability of a specific mAb against the human pTα could be a valuable T-ALL diagnosis and prognosis tool, although the reduced membrane levels of this complex make its detection challenging [175]. Collectively, the aforementioned studies suggest a key role for NOTCH1-pre-TCR cross-talk in T-ALL pathogenesis, indicating that the signal intensity provided through either the pre-TCR or NOTCH1 could modulate both the latency and the aggressiveness of the disease. Therefore, pre-TCR could be envisaged as a key biomarker in NOTCH1-dependent T-ALL, thus opening new avenues for the design of combinatorial therapies targeting both signaling pathways.

### 3.6. Metabolic Regulators

The regulation of cellular metabolism by NOTCH has emerged in recent years as an exciting area of research. Although still quite unexplored, strong evidence supports the idea that NOTCH1 signaling promotes T-ALL growth by inducing the activation of anabolic pathways, including glycolysis, glutaminolysis, nucleotide biosynthesis, amino acids metabolism, ribosome biogenesis and protein activation [105,176,177,178]. On the contrary, NOTCH1 signaling inactivation switches metabolism to catabolic pathways, i.e., increased autophagy, ubiquitination and proteasomal degradation, and reduction in glycolytic and glutaminolytic flux [105]. To date, the best-known metabolic routes cooperating with oncogenic NOTCH1 signaling are MYC, PI3K/AKT, and RAS, and they will be discussed below. In addition, very recent studies have uncovered novel players involved in the metabolic reprogramming caused by aberrant NOTCH1 signals, such as the oxidative phosphorylation regulator mitochondrial complex I inhibitor [178] and the nucleotide biosynthesis regulator ubiquitin protein ligase E3 component N-recognin 7 (UBR7) [177]. Given that metabolic reprogramming is an essential mechanism of cancer cell outgrowth and dissemination [179], understanding the multiple levels of interconnections between NOTCH1 and metabolic pathways may help to delineate novel strategies to tackle exacerbated metabolic demand in T-ALL. Several metabolic regulators with a prominent impact on T-ALL oncogenesis will be discussed below.

#### 3.6.1. MYC

MYC is a master regulator involved in many cellular metabolism processes. As a transcriptional regulator, MYC displays the essential ability to directly couple sensing and acquisition of nutrients to molecular pathways that control the growth and proliferation of healthy cells and whose aberrant activation confers growth factor independency to cancer cells [180]. During T-cell development, MYC expression is upregulated in DN3 and DN4 stages [181,182], coincident with NOTCH1 and pre-TCR signaling [183], decreasing later in DP and SP stages. Accordingly, a deficient *Myc* expression in murine T-cell progenitors was shown to result in the impaired expansion of TCRβ rearranged cells at DN3-DN4 stages and caused a dramatic loss of DP cells, indicating that MYC function is essential for intrathymic T-cell generation [184]. Depletion of *Myc* later in development, using *CD4-Cre* conditional mice, demonstrated an essential MYC function in the differentiation of effector and memory T cells [185] associated with TCR- and interleukin-2 (IL-2)-dependent MYC regulation [186].

According to its expression in normal T cells, MYC is also broadly expressed in T-ALL [187], displaying a prominent role in both cell proliferation and survival [108,188,189]. Deregulated MYC expression in T-ALLs is partially attributed to the presence of chromosomal translocations in 6% of T-ALL patients, encompassing the *MYC* locus and TCR genes or other partners, such as *CDK6* [190]. The oncogenicity of MYC deregulation in T cells has been demonstrated in zebrafish [191,192] and mouse models [193], in which transgenic *MYC* expression was probed sufficiently to induce T-cell lymphomagenesis. However, the inability of sporadic MYC activation to initiate a *bonafide* murine T-ALL questioned the action of MYC deregulation as a driving oncogene in T-ALL [194]. Despite these findings, several studies have demonstrated the dependency of T-ALL on aberrant MYC signals. Downregulation of MYC activity either by shRNA or by JQ1/CPI203 treatment reduced LIC frequency, impaired T-ALL progression and increased overall survival in murine T-ALL models [188,189,195].

Importantly, both in normal and in neoplastic T cells, MYC expression was directly regulated by NOTCH1 [196]. Although initial experiments failed to determine an active role of NOTCH1 on *MYC* promoter regulation, deeper chromatin immunoprecipitation (ChIP) analysis in T-ALL led to the identification of a T-cell-specific distal genomic region located >1 Mb 3′ of human and murine *MYC* locus strongly occupied by NOTCH transcription complexes, which physically interacts with the regulatory sequences in the *MYC* proximal promoter [108,197]. Recurrent somatic focal duplications of this NOTCH/MYC enhancer (*N-Me*) region were found in ~5% of T-ALL patients [108]. Supporting the oncogenic role of the *N-Me* region, leukemia development is impaired in mice transplanted with *N-Me*-deficient BM progenitors transduced with an oncogenic NOTCH1 allele (ΔE-NOTCH1). *N-Me* deficiency also leads to a reduction in thymus size and cellularity, mainly caused by a DN3-stage developmental block, alongside the consequent reduction in DP and SP cells [108], a phenotype similar to that described for MYC-deficient T cells [184]. These results further confirm the crucial function of MYC during β-selection and T-cell development. Additional studies by Ferrando’s group uncovered the role of the transcription factor GATA3 in regulating chromatin accessibility at the *N-Me*, both in thymocytes and in T-ALL [198], and suggested an epistatic function of GATA3 and *N-Me* over NOTCH1 signaling during T-cell transformation. Moreover, FBXW7- [195] and PTEN-dependent [108,192,199] posttranscriptional regulation of MYC has been described in T-ALL.

Taken together, available data indicate that MYC is a suitable therapeutic target, not only for T-ALL patients with *NOTCH1/FBXW7* mutations but also for NOTCH-independent T-ALLs where additional signaling pathways, including PTEN and PI3K/AKT (see below), are deregulated. Nonetheless, since MYC represents a challenging druggable protein because of its multiple levels of regulation, additional studies addressing MYC posttranscriptional modifications and conformational changes would be helpful to obtain more effective and affordable inhibitors to be used in the clinic.

#### 3.6.2. PI3K/AKT/mTOR

PI3K/AKT signaling has been largely associated with cellular metabolism by regulation of different pathways, including expression of nutrient transporters, biosynthesis of macromolecules, maintenance of redox balance and glucose metabolism [200]. Under physiological conditions, the PI3K/AKT pathway is activated by growth factors, insulin or cytokines, resulting in binding to phosphorylated tyrosine residues within activated receptors and adaptor proteins. At the plasma membrane, PI3K activation induces phosphatidylinositol 4,5-bisphosphate (PIP_2_) phosphorylation and generation of phosphatidylinositol 3,4,5-trisphosphate (PIP_3_), which accumulates and serves as a docking site for pleckstrin homology (PH) domain-containing proteins, such as AKT. Subsequent phosphorylation events lead to full AKT activation, which in turn activates several metabolic regulators, including mTOR, glycogen synthase kinase 3 (GSK3) and forkhead box protein O (FOXO), and contribute to transcriptional regulation of key effectors of metabolic pathways. Conversely, PI3K signaling is counteracted by the action of phosphatase and tensin homolog (PTEN), which dephosphorylates PIP_3_ to restore PIP_2_ levels. Activation of AKT has also been shown to favor glucose uptake by modulation of glucose transporter-1 (GLUT1) trafficking to the cell surface [201,202,203] and by activation of glycolytic enzymes, such as hexokinase 2 (HK2) [204] and 6-phosphofructo-2-kinase/fructose-2,6- biphosphatase (PFKFB2) [205].

A central role has been described for the NOTCH1/PI3K/AKT axis in regulating T-cell metabolism. NOTCH1 withdrawal in DN pre-T cells causes cellular atrophy and programmed cell death, in addition to reduced GLUT1 expression and glycolytic rate, indicating that NOTCH1 signals might promote glucose metabolism in developing T cells [165]. The trophic action of NOTCH1 and pre-TCR at the β-selection checkpoint was shown to be assisted by IL-7R signaling, which induced CD71 (transferrin receptor) and CD98 (neutral amino acid transporter) expression, and collaborated with NOTCH1 in DN4 cell self-renewal [206]. Remarkably, PI3K/AKT activation downstream of IL-7R was shown to induce the expression of GLUT1 in T-ALL cells and stimulated glucose uptake [201], pointing to IL-7R as an essential intermediate in the NOTCH/PI3K/AKT pathway.

Also, NOTCH1 directly regulates PI3K/AKT pathway activity by modulating PTEN expression, as demonstrated by comparative global gene expression analysis showing a specific reduction of PTEN expression in GSI-resistant T-ALLs, as compared with GSI-sensitive NOTCH1-dependent T-ALLs [176]. Molecular studies established that NOTCH1 inhibits PTEN expression via HES1, a transcriptional repressor that mediates many NOTCH1-induced transcriptional programs. Genetic abnormalities, including point mutations, gene deletions, micro-deletions in the *PTEN* gene, and posttranslational inactivation of the PTEN protein, are frequently found in T-ALL [207,208]. *PTEN* loss-of-function, leading to constitutive activation of PI3K/AKT-dependent growth signals in T-ALL, might bypass NOTCH1 requirements to maintain cell survival, supporting the generation of T-ALL cells resistant to NOTCH1 inhibition [176]. Seminal studies by Herranz and colleagues using a NOTCH1-dependent TALL mouse model with inducible loss of *Pten* revealed that metabolic changes were responsible for the resistance to GSI of *Pten* loss-of-function T-ALLs [105]. They showed that murine T-ALL cells induced by active NOTCH1 were highly dependent on glutaminolysis as a source of carbon. Thus, concomitant administration of GSI and glutaminase (GLS) inhibitors showed synergistic anti-leukemic effects against *Pten*-positive T-ALLs, while *Pten*-negative leukemias were resistant not only to GSI but also to the combination of both GSI and GLS inhibitors. Mechanistically, *Pten*-deleted T-ALLs displayed glycolysis hyperactivation and increased glucose-derived carbon input to the TCA cycle, rendering cells more resistant to glutaminolysis inhibition [105]. Therefore, metabolic unbalance caused by NOTCH1 inhibition in T-ALLs forces cells to rely on catabolic pathways to obtain essential metabolites for cell growth, this finding opening new therapeutic avenues to overcome GSI resistance in T-ALL patients.

Additional studies have uncovered other complex cross-regulation mechanisms between NOTCH1 and PI3K/AKT pathways. It was shown that inhibition of AKT using pharmacological inhibitors decreases NOTCH1 protein levels [209] by inducing NOTCH1 lysosomal degradation through C-CBL (an E3 ubiquitin ligase)-mediated monoubiquitination [210]. Besides regulating PI3K/AKT activity through PTEN inhibition, NOTCH1 can also regulate the PI3K/AKT pathway by modulating PP2A-dependent AKT phosphorylation [211]. Moreover, NOTCH1 regulates the phosphorylation of multiple signaling proteins of the mTOR pathway through a PI3K/AKT-independent but MYC-dependent mechanism [212]. mTOR is an essential regulator of T-ALL cell growth, and its inhibition has been demonstrated to have anti-leukemic effects either alone or in combination with other drugs [106,213,214,215,216,217,218,219]. Notably, dual inhibition of NOTCH1 and mTOR signaling was proved to act synergistically by blocking T-ALL proliferation in vitro [106]. Dual NOTCH1 and mTOR targeting were also beneficial in preclinical models in which human T-ALL xenotransplanted mice treated with a combination of GSI and rapamycin showed increased survival compared to mice treated with GSI alone [107].

In conclusion, PI3K/AKT/mTOR represents a major signaling pathway to be targeted in combination with NOTCH1 inhibitors, especially in T-ALL patients harboring PTEN inactivation where constitutive activation of PI3K/AKT signaling may lead to GSI resistance.

#### 3.6.3. KRAS

KRAS belongs to the RAS protein family of small GTPases that cycle between an active (GTP-bound) and an inactive (GDP-bound) state by means of regulation by guanine-nucleotide-exchange factors (GEFs) and GTPase-activating proteins (GAP)s. Commonly, stimulation of receptor tyrosine kinases leads to recruitment of protein complexes that activate RAS, which subsequently recruits and activates several effectors, including MAPK (RAF/MEK/ERK) and PI3K/AKT, regulating multiple cellular processes such as proliferation, differentiation, survival and cytoskeleton reorganization. KRAS activation has been shown to balance metabolism towards anabolic pathways through metabolic rewiring that involves the up-regulation of enzymes that mediate glycolysis, together with amino acids, fatty acids and nucleotide biosynthesis [220]. Furthermore, KRAS mutations stimulate scavenger pathways, such as macropinocytosis and autophagy, to sustain cancer cell survival under nutrient deprivation [220].

Although KRAS is one of the most frequent mutated oncogenes in cancer, mutations in KRAS appear at low frequency in ALL patients at diagnosis [221]. However, the frequency of KRAS mutations increases in relapses, suggesting either therapy-induced clonal selection or secondarily-acquired mutations [222,223]. Experimentally, conditional hematopoietic expression of an active *Kras^G12D^* mutant was shown to induce a fatal monocytic myeloproliferative disorder (MPD) in first-recipient mice that was not transplantable [224]. Instead, a highly penetrant aggressive T-cell disease developed in secondary recipient mice, which showed characteristic extrathymic immature DP CD44^+^ leukemic cells [55,225]. Mutational screening uncovered the presence of secondarily acquired NOTCH1 mutations, located in the PEST domain and/or the overexpression of NOTCH1, in 100% of mice, which correlated with a high level of transplantability of these leukemias [225]. On the contrary, preleukemic cells at early stages of leukemia development lacked NOTCH1 mutations and were unable to generate leukemia in secondary recipients [225], indicating that the acquisition of NOTCH1 mutations increases KRAS-induced T-ALL aggressiveness. Accordingly, oncogenic KRAS was shown to induce a preleukemic state in HPCs that requires secondarily acquired NOTCH1 mutations to fatally induce the transformation of developing T cells into T-ALL [224]. Secondary acquisition of NOTCH1 mutations was also confirmed in T-ALLs arising in an alternative *Kras^G12D^*–induced T-ALL mouse model [226]. Notably, survival and proliferation of T-ALL cell lines derived from those mice showed a great dependency on both NOTCH1 and Ras pathway activation [55,226]. Cooperation between KRAS and NOTCH in T-ALL was directly demonstrated by Pear and colleagues [54] in studies showing that *NOTCH1* GOF alleles found in T-ALL patients efficiently generate leukemia in mice when overexpressed in transplanted BM HPCs expressing *Kras^G12D^* but not in wild-type BM HPCs.

Collectively, the above results point towards activated KRAS as an important oncogenic pathway cooperating with NOTCH1 in T-ALL, which could potentially be targeted by pharmacological inhibitors. Despite KRAS has been considered a virtually undruggable target for years, several RAS signaling inhibitors have been recently developed [227], and many are currently being tested in clinical trials for different cancers. Therefore, numerous T-ALL patients could still benefit from these novel compounds targeting RAS pathway activation downstream of deregulated signaling from cytokine or growth factor receptors (i.e., IL-7R, IGFR1) frequently found in T-ALL (Figure 3).

### 3.7. Cell Cycle Regulators

Deregulation of cell cycle progression is widely associated with oncogenic transformation [179] and has a prominent role in T-ALL pathogenesis [43]. D-type cyclins are the ultimate recipients of mitogenic and oncogenic signals. Of them, CYCLIN D3 is required during T-cell development for p56^LCK^-mediated pre-TCR dependent expansion at the β-selection checkpoint and thus for the proliferation of DN4 and ISP thymocytes and generation of DP thymocytes [228]. Notably, *Ccnd3* deficiency in mice impairs the generation of T-ALL induced by NOTCH1, highlighting the relevance of CYCLIN D3-dependent cell cycle progression in NOTCH1-mediated oncogenesis. Supporting a direct role of oncogenic NOTCH1 signaling in G1-S cell cycle progression, *CCND3* was confirmed as a transcriptional target of NOTCH1 in T cells [109]. NOTCH function also controls the expression of cyclin-dependent kinases CDK4 and CDK6 that associate with D-type cyclins and regulate retinoblastoma (RB) phosphorylation [109]. Consequently, NOTCH1-induced T-ALL pathogenesis and/or disease maintenance and progression were shown to be dependent on *CCND3*, *CDK4* and *CDK6* gene expression [109,110] and on down-regulation of the cyclin-dependent kinase inhibitors *CDKN2D* (p19INK4d) and *CDKN1B* (p27Kip1) [229]. In this regard, NOTCH1 was shown to induce transcription of the S-phase kinase-associated protein 2 (SKP2), a negative regulator of CDKN1A and CDKN1B cell cycle inhibitors [230]. Importantly, both CYCLIN D3 and CDK4 are highly overexpressed in NOTCH-dependent T-cell tumors [229]. Moreover, activating mutations in NOTCH1 frequently co-occur with loss of the *CDKN2A* locus, which encodes the tumor suppressors p16INK4A and p14ARF, and *CDKN2A* deletions are found in most (>70%) T-ALLs [40,43,231,232], while 15% and 12% of T-ALLs show chromosomal deletions involving the *RB1* locus [232,233], and the *CDKN1B* gene [234], respectively.

Collectively, the reported cross-talk between oncogenic NOTCH1 signaling and cell cycle regulators highlights the need to design accurate combined therapies targeting cell cycle deregulation in NOTCH1-dependent T-ALLs.

### 3.8. Adhesion/Chemotactic Receptors

#### 3.8.1. Chemokine Receptors

Activation of chemokine receptor CXCR4 by its ligand CXCL12 (stromal cell-derived factor α, SDF1-α) is known to mediate critical interactions of HPCs with the BM niche that regulate BM homing and engraftment [235,236]. Importantly, CXCR4-CXCL12 signaling supports as well the interaction of leukemic cells with the BM microenvironment, regulating the survival and progression of NOTCH1-induced mouse and human T-ALL cells and the expansion of human T-ALL xenografts mediated by LICs [92,111,112,237]. Confirming that LIC activity is dependent on CXCR4 expression, downregulation of CXCR4 in NICD-induced T-ALLs was shown to promote apoptosis, decrease proliferation and reduce BM leukemic burden in vivo [237]. Also, mice with a specific deletion of the *Cxcl12* CXCR4 ligand in vascular endothelial cells revealed reduced expansion of NOTCH1-dependent T-ALL cells in BM and peripheral organs [111]. These findings concur with the observation that overexpression of CXCR4 in hematological tumors correlates with disease progression [238,239], pointing to CXCR4 as an important player in the T-cell oncogenic program induced by NOTCH1. Regarding possible crosstalk between NOTCH1 signaling and CXCR4 function, NOTCH1 does not seem to directly control CXCR4 expression [92,113,114]. However, NOTCH3, another member of the NOTCH receptor family with an established T-cell oncogenic capacity [240], has been shown to upregulate CXCR4 expression in mouse T-ALL cells by inhibiting receptor internalization through modulation of ARRESTIN-1β localization and function [241]. Additional regulatory mechanisms have been described using a mouse model of de novo generation of human T-ALL, where NOTCH1 activation was shown to increase the functional competence of CXCR4 in leukemic cells without altering protein expression levels [92]. Therefore, NOTCH1 signaling may promote the interaction of leukemic cells with the supportive BM niche by regulating the activity of the CXCR4/CXCL12 signaling axis.

Of note, CXCR4’s function is not only involved in the BM engraftment of T-ALL cells but also seems to guide them to the colonization of different organs [242], including the central nervous system (CNS). Migration of T-ALL cells to the CNS has also been shown to be dependent on the expression of the C-C- chemokine receptor 7 (CCR7) in oncogenic NOTCH1 mouse models [113]. In this case, constitutively active NOTCH1 was found to upregulate *Ccr7* gene expression in murine hematopoietic progenitors, and CCR7 expression in T-ALL cell lines was sensitive to GSI treatment, supporting a direct role of NOTCH1 activation in the regulation of CCR7 expression. In vivo, *Ccr7* deletion showed no effect on NICD-induced murine T-ALL generation or T-ALL cell infiltration of most tissues but specifically impaired CNS infiltration [112]. In contrast, deletion of *Ccr7* failed to impair CNS infiltration by B-ALLs, thus pointing to a specific function for CCR7 in NOTCH1-dependent T-ALL.

The relevance of the above findings is supported by results derived from primary human T-ALLs, showing that expression of high levels of mRNA of both CCR7 and CXCR4 correlated with a high risk of CNS involvement in non-relapsed patients [243]. Collectively, these data highlight the potential of CXCR4 and CCR7 chemokine receptors as powerful therapeutic targets for the treatment of T-ALL patients with NOTCH1 activating mutations, especially for those with CNS involvement (Figure 3). Other chemokine receptors, including CCR5 and CCR9, which regulate T-ALL proliferation and migration *in vitro*, are known to be transcriptionally regulated by NOTCH1 and could be likewise considered optimal T-ALL therapeutic targets. While the functional relevance of CCR5 in T-ALL progression has yet to be determined in vivo [114], very recent studies have demonstrated that targeting of CCR9, which is expressed in >70% of cases of T-ALL, has potent anti-leukemic activity in vivo in human T-ALL xenografts and could be a highly effective therapeutic strategy for T-ALL [244,245] (Figure 3).

#### 3.8.2. CD44 Adhesion Receptor

Adhesion molecules that mediate the interaction of T-ALL cells with their specific niches are key contributors to leukemia progression. Of special relevance is CD44, a transmembrane glycoprotein with several isoforms resulting from alternative splicing, whose main ligand, hyaluronan, is present in the extracellular matrix [246]. CD44 plays a key role as a physiological mediator of HPC anchoring within the BM niche during hematopoiesis and also mediates BM engraftment of aberrant hematopoietic cells [247]. Specifically, several studies have highlighted the clinical significance of CD44 as a tumor-initiating marker of chronic (CML) [248] and acute (AML) myeloid leukemias [249], and CD44 expression levels have also been associated with LIC activity in murine models of T-ALL, were a link between NOTCH signaling and CD44 expression was observed [250]. More recently, upregulation of CD44 has been proposed as one of the initial events encompassing human T-ALL generation in mice transplanted with human HPCs expressing NICD, in which CD44 was shown to support BM engraftment and expansion and further progression of human preleukemic cells [92]. This study also demonstrated that *CD44* is a direct NOTCH1 target, whose transcription is regulated by MAML1/CSL-binding to its proximal promoter, but also through dynamic interactions with a superenhancer, which is a mechanism shared with other NOTCH1 targets involved in development and cancer, such as IL7R [30,31]. Also, NOTCH1-induced upregulation of CD44 was essential for NOTCH1-dependent human T-ALL pathogenesis since treatment with anti-CD44 antibodies reduced BM engraftment of NICD^+^ preleukemic cells, and eradicated disease progression of established T-ALL patient-derived xenografts [92].

Supporting a clinical relevance for CD44 targeting in T-ALL, aberrant expression of CD44 has been detected in different T-ALL subtypes, and abnormally high expression levels of CD44 have been associated with organ infiltration in T-ALL patients [251] and with T-ALL chemoresistance [252]. These results suggest that expression of CD44 may provide LICs with a retention capacity to specific niches within the BM, where they could be kept protected from treatment and contribute later to leukemia relapses. In line with this hypothesis, CD44 expression has been identified as a predictor of T-ALL relapse [253]. Therefore, CD44 may be a useful marker in T-ALL patients with NOTCH1 mutations that could help in the prognosis and evaluation of relapse risk and hence can be considered a potential target for future combined therapies (Figure 3).

## 4. Concluding Remarks and Future Perspectives

Although intensive chemotherapy treatments have greatly improved T-ALL patient survival in recent years, relapses and refractories still have a disheartening prognosis. Since the discovery that constitutive activation of NOTCH1 signaling is a major player in the T-cell oncogenic process, results from ongoing anti-NOTCH1 clinical trials are expected to offer T-ALL patients a more specific, effective and targeted treatment to improve their life expectancy. However, most of the naturally-found NOTCH1 mutants in T-ALL patients are not capable of inducing complete T-cell transformation in animal models, which raised the notion that other players might be acting in concert with NOTCH during the T-ALL oncogenic process. As expected, the majority of these players comprise signaling pathways that have been known for a long time as essential regulators of cell metabolism, proliferation and survival, which are deregulated in many types of cancers. This scenario represents a great opportunity for future therapeutic developments for T-ALL patients since numerous modulators of these pro-oncogenic signaling pathways have been approved for clinical use for other pathologies. Other oncogenic collaborators of NOTCH1, such as RNA regulators, have been revealed as potential targets of therapeutic intervention in recent years, and their potential clinical relevance needs to be confirmed. Therefore, innovative technologies focusing on the identification of the deregulated pathways collaborating with NOTCH1 in T-ALL patients, in combination with human T-ALL in vivo models that provide proof-of-concept of the efficacy and safety of novel combinational therapies targeting both NOTCH1 and associated oncogenic collaborators, will open new avenues for developing in the near future efficacious and safe targeting therapies for T-ALL patients.

## Figures and Tables

**Figure 1 ijms-24-01383-f001:**
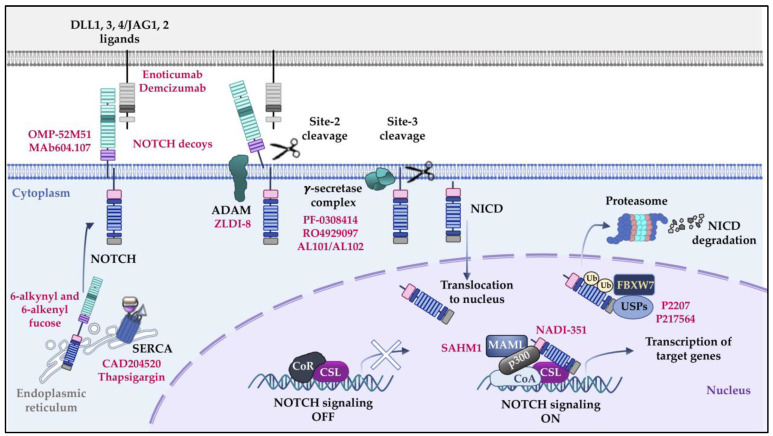
NOTCH signaling and targeting strategies. Interaction of NOTCH receptors with DLL/JAG ligands exposes site-2 located within the HD domain of extracellular NOTCH, allowing proteolytic cleavage by ADAM metalloproteases. Sequential cleavage by a γ-secretase complex at the transmembrane region (site-3) of the receptor releases the NOTCH intracellular domain (NICD) that translocates to the nucleus and binds to the transcription factor CSL. In the absence of NOTCH signaling (NOTCH signaling OFF), CSL actively represses transcription of NOTCH target genes owing to interaction with corepressors (CoR). Upon NOTCH activation (NOTCH signaling ON), NICD recruits a transcriptional complex (including MAML, p300 and CoA) that displaces CoR from CSL, allowing CSL to become transcriptionally active and to induce NOTCH target gene transcription. Thereafter, FBXW7-dependent ubiquitination and subsequent proteasomal degradation of NICD lead to NOTCH signaling shutdown. NOTCH signaling can be efficiently inhibited by molecular targeting at different levels (red) by: (1) blocking receptor-ligand interactions with monoclonal antibodies (mAbs) [i.e., anti-NOTCH1 (OMP-52M51, Mab604.107) and anti-DLL4 (Enoticumab, Demcizumab) mAbs], or with NOTCH decoys; (2) inhibiting receptor proteolytic cleavage with ADAM inhibitors (ZLDI-8) or γ-secretase inhibitors (i.e., PF-0308414, RO4929097, AL101/AL102); (3) blocking NOTCH transcriptional complex activation with MAML1 inhibitors (SAHM1) or NOTCH ternary complex inhibitors (NADI-351); (4) favoring NOTCH degradation with USP7 inhibitors (P2207, P217564); or (5) interfering with NOTCH receptor maturation with POFUT1 inhibitors (6-alkynyl and 6-alkenyl fucose) and SERCA inhibitors (CAD204520, Thapsigargin).

**Figure 2 ijms-24-01383-f002:**
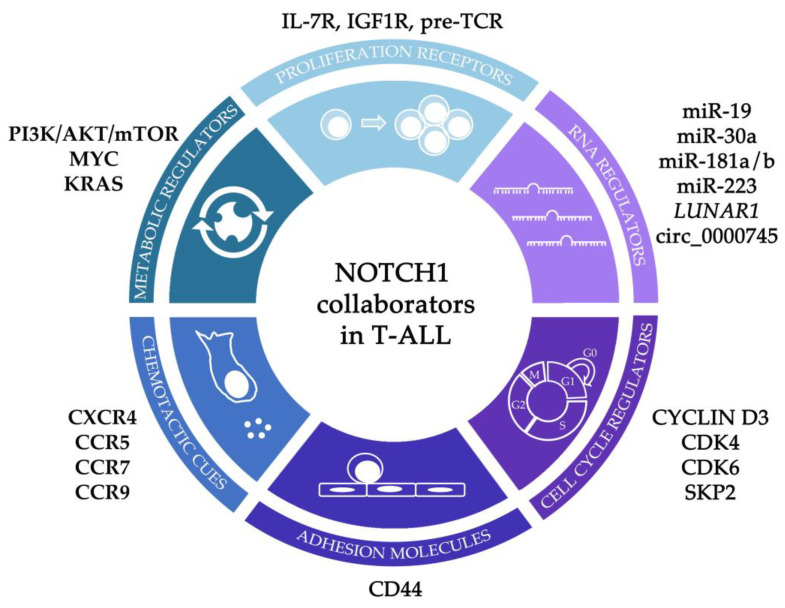
NOTCH1 pathway collaborators contributing to T-ALL pathogenesis.

**Figure 3 ijms-24-01383-f003:**
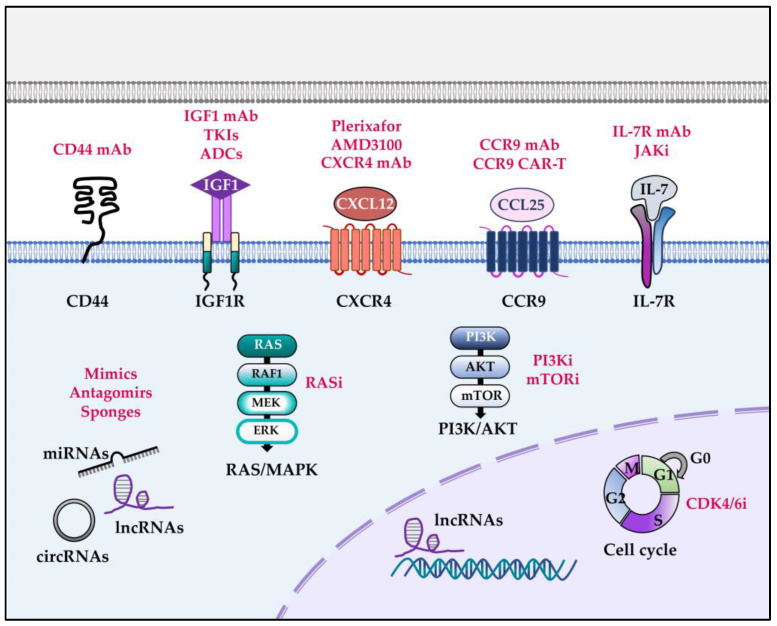
Preclinical therapies targeting NOTCH1 collaboration partners in T-ALL pathogenesis. Therapeutic approaches (black) specifically targeting signaling pathways and molecular regulators (red) that collaborate with NOTCH1 in T-ALL pathogenesis proved successful in preclinical mouse models. ADCs (antibody-drug conjugates); CAR-T (chimeric antigen receptor T cells); i (inhibitor); mAb (monoclonal antibody); TKIs (tyrosine kinase inhibitors).

**Table 1 ijms-24-01383-t001:** NOTCH1 pathway targeting strategies.

Inhibitor	Target	Mechanism	References
P2207, P217564	USP7	Small molecule inhibiting USP7-dependent NOTCH1 deubiquitination	[6,7]
PF-0308414	γ-secretase complex	Small molecule inhibiting γ-secretase dependent NOTCH1 cleavage	[64,68,70]
RO4929097	γ-secretase complex	Small molecule inhibiting γ-secretase dependent NOTCH1 cleavage	[64,68]
AL101/AL102	γ-secretase complex	Small molecule inhibiting γ-secretase dependent NOTCH1 cleavage	[64,68,69]
OMP-52M51 (Brontictuzumab)	NOTCH1	Antibody interfering with NOTCH1-ligand interaction	[71]
Mab604.107	NOTCH1	Antibody interfering with NOTCH1-ligand interaction	[72]
Demcizumab	DLL4	Antibody interfering with NOTCH-DLL4 ligand interaction	[73]
Enoticumab	DLL4	Antibody interfering with NOTCH-DLL4 ligand interaction	[74]
ZLDI-8	ADAM17	Small molecule inhibiting ADAM17-dependent NOTCH1 cleavage	[79]
NOTCH decoys	NOTCH1	Soluble inhibitors interfering with NOTCH1-ligand interaction	[80]
SAHM	MAML1	Stapled α-helical peptides interfering with MAML1 binding to NOTCH transcriptional complex	[81]
NADI-351	NOTCH Ternary complex (NTC)	Small molecule disrupting NTC	[82]
Thapsigargin	SERCA	Small molecule inhibiting SERCA activity	[83]
CAD204520	SERCA	Small molecule inhibiting SERCA activity	[84]
6-alkynyl and 6-alkenyl fucose	POFUT1	L-fucose analogs disrupting NOTCH1-DELTA-ligand interactions	[85]

## Data Availability

Not applicable.

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
