# Peer review of "Notch Partners in the Long Journey of T-ALL Pathogenesis"

_ijms, 2023, doi:10.3390/ijms24021383_

Round 1
Reviewer 1 Report
Toribio and Gonzalez-Garcia nicely review the current literature of the contributing factors in the pathogenesis leading to T-cell acute lymphoblastic leukemia. Overall, this is a convincing and comprehensive review. I still have a few comments/suggestions that should be addressed before publication:
1) Figure-2 is a repetition of Fig.1. Please integrate the most important information of Fig.2 into Fig.1. Instead of Fig-2, I suggest a comprehensive table with all the useful terms.
2) Page-7: Non-coding RNAs are not really signaling pathways. I suggest to do a seperate section (i.e. section-3: RNA modulators of....; section-4: Signaling pathways)
3) Page-7: Concerning Figure-3: Relevant Pathways
These are not ‘relevant pathways’. For me, this implies signal transduction pathways.
I would name them Notch pathway modulators (or similar).
4) Page-2: Please also add USP-10 as a major DUP (ubiquitin specific protease)
Reference: Lim et al., 2019, Science, PMID: 30975888
USP10 biochemically interacts with NICD1 in T-ALL and this is important for Notch target gene expression (Ferrante et al., 2022; Cell Death and Disease PMID: 35821235).
Most likely there are several USPs important for NICD1. (The same will be true for E3 ligases).
Minor points:
5) Please if human use all capital letters and if mouse only the first letter is a capital letter.
Page-2: Several cellular- and context-dependent NOTCH transcriptional 47 targets have been described so far, including Hes1-5, Hey1,2, Ptcra, Myc, Dtx1, IL7R, CD44 48 and Runx1 [3]
Concerning Notch target genes: When human, please use capital letter; when mouse, small letters. ....HES1-5, HEY-1 and -2
6) In vitro and in vivo and de novo (italics):
Please correct throughout the manuscript. In vitro, in vivo and de novo should be written in italics.
Author Response
Point by point reply to Reviewer 1’s Comments
- Figure-2 is a repetition of Fig.1. Please integrate the most important information of Fig.2 into Fig.1. Instead of Fig-2, I suggest a comprehensive table with all the useful terms.
We thank the Reviewer for his/her suggestion. Accordingly, the revised manuscript now includes a new Figure 1 that integrates the most important information from old Figure 2, which has been removed. As suggested, we also include a new Table 1, containing useful terms and information on the mechanism of action of the different inhibitors targeting NOTCH1 signaling depicted in new Figure 1.
2) Page-7: Non-coding RNAs are not really signaling pathways. I suggest to do a seperate section (i.e. section-3: RNA modulators of....; section-4: Signaling pathways)
We agree with the Reviewer that non-coding RNAs are not really signaling pathways. Therefore, for the sake of clarity, the term “signaling pathway” has been omitted from the section 3 heading, which is now more general and refers to molecular collaborators of NOTCH1 signaling in T-ALL pathogenesis, including several separate subsections referring to RNA modulators, survival/proliferation signaling receptors, metabolic regulators, cell cycle regulators and adhesion/chemotactic receptors.
3) Page-7: Concerning Figure-3: Relevant Pathways
These are not ‘relevant pathways’. For me, this implies signal transduction pathways.
I would name them Notch pathway modulators (or similar).
Following the Reviewer’s suggestion, the term “relevant pathways” has been removed from the legend to new Figure 2 (old Figure 3), and is now replaced by “NOTCH1 pathway collaborators”
4) Page-2: Please also add USP-10 as a major DUP (ubiquitin specific protease)
Reference: Lim et al., 2019, Science, PMID: 30975888
USP10 biochemically interacts with NICD1 in T-ALL and this is important for Notch target gene expression (Ferrante et al., 2022; Cell Death and Disease PMID: 35821235).
Most likely there are several USPs important for NICD1. (The same will be true for E3 ligases).
According to the Reviewer ‘s suggestion, we have added new information related to USP-10 (and USP7) deubiquitinases, and Su(dX), NEDD4 and MIB1/2 E3 ligases, together with the corresponding references in Section 1 (page 2)
Minor points:
5) Please if human use all capital letters and if mouse only the first letter is a capital letter.
Page-2: Several cellular- and context-dependent NOTCH transcriptional 47 targets have been described so far, including Hes1-5, Hey1,2, Ptcra, Myc, Dtx1, IL7R, CD44 48 and Runx1 [3]
Concerning Notch target genes: When human, please use capital letter; when mouse, small letters. ....HES1-5, HEY-1 and -2
According to the Reviewer ‘s recommendation we have used capital letters for human terms and small letters for mouse terms throughout the revised manuscript
6) In vitro and in vivo and de novo (italics):
Please correct throughout the manuscript. In vitro, in vivo and de novo should be written in italics.
In vitro, in vivo and de novo are now written in italics throughout the revised manuscript.

Reviewer 2 Report
Authors have presented an interesting review article entitled “Notch partners in the long journey of T-ALL pathogenesis”. Overall, the paper is well written and highlighted the role of Notch signaling and their key mediators in T-ALL pathogenesis. In my opinion, the manuscript can be published in this journal, after the authors have addressed the following comments and questions:
· Section 1 is quite lengthy and unnecessary information incorporated which is previously described by many others. In my opinion authors should be specific in this section, and try to concise this section only with relevant information related to tumor promoting potential of Notch signaling.
· Author have included lot of content in section and subsections; however, these sections and subsections are not properly linked to each other. Thus, authors are suggested to link up the content.
· Section 2.1, authors have provided the information related to Notch1 only while heading represent NOTCH pathway targeting strategies. Which one is correct?
· In table 1, authors are suggested to add abbreviations in footnote.
· In section 3 authors have described the main targets of Notch1 but cross linkage with several signaling pathway is limited. Please correct it accordingly.
· If possible, try to add some information regarding clinical studies related to notch inhibitors separately.
· Abbreviations should be clearly stated throughout the manuscript.
Author Response
Point by point reply to Reviewer 2 ‘s Comments
Comments and Suggestions for Authors
Authors have presented an interesting review article entitled “Notch partners in the long journey of T-ALL pathogenesis”. Overall, the paper is well written and highlighted the role of Notch signaling and their key mediators in T-ALL pathogenesis. In my opinion, the manuscript can be published in this journal, after the authors have addressed the following comments and questions:
- 1) Section 1 is quite lengthy and unnecessary information incorporated which is previously described by many others. In my opinion authors should be specific in this section, and try to concise this section only with relevant information related to tumor promoting potential of Notch signaling.
We thank the Reviewer for his/her suggestion. Accordingly, information in section 1 regarding the role of NOTCH1 in T-cell development has been shortened in the revised manuscript, and only introductory information related to developmental checkpoints and NOTCH1 downstream effectors involved in NOTCH1-mediated T-ALL pathogenesis is now included.
- 2) Author have included lot of content in section and subsections; however, these sections and subsections are not properly linked to each other. Thus, authors are suggested to link up the content.
According to the Reviewer’s concern, we have modified section 3 in the revised manuscript, in order to linked up its content. Thus, section 3 heading has been changed and the term “signaling pathways” has been replaced by “molecular collaborators” (of NOTCH1 signaling in T-ALL pathogenesis), to accommodate distinct subgroups of oncogenic collaborators in separate subsections (metabolic and cell cycle regulators, proliferation and chemokine/adhesion receptors, and also RNA modulators, which as pointed out by Reviewer 1, are not really signaling pathways). These subsections are linked each other by an introductory paragraph at the end of section 2 (page 5) and the beginning of section 3 (page 6) and also in new Figure 2 (old Figure 3). Also, for the sake of clarity, sub-subsections have been eliminated in the revised manuscript.
- 3) Section 2.1, authors have provided the information related to Notch1 only while heading represent NOTCH pathway targeting strategies. Which one is correct?
According to the Reviewer concern, section 2.1 heading has been modified in the revised manuscript and NOTCH pathway has been substituted by NOTCH1 pathway, given that no information is yet available regarding the efficacy of the mentioned inhibitors over activation of NOTCH receptors distinct than NOTCH1.
- 4) In table 1, authors are suggested to add abbreviations in footnote.
We thank the Reviewer for his/her suggestion. Abbreviations have now been added in the footnote of old Table 1 (new Table2).
- 5) In section 3 authors have described the main targets of Notch1 but cross linkage with several signaling pathway is limited. Please correct it accordingly.
We agree with Reviewer 2 that using terms such as “cross-talk” or “cross-regulation” may not be adequate, as limited data on cross linkage with other pathways is provided. Therefore, for the sake of clarity, such terms have been eliminated from section 3 and throughout the revised manuscript. Also, the term “signaling pathway” has been omitted from the section 3 heading, which is now more general and refers to molecular collaborators of NOTCH1 signaling in T-ALL pathogenesis, including several separate subsections, referring to RNA modulators, survival/proliferation signaling receptors, metabolic regulators, cell cycle regulators and adhesion/chemotactic receptors, as summarized in new Figure 2.
- 5) If possible, try to add some information regarding clinical studies related to notch inhibitors separately.
Following Reviewer 2 ‘ s recommendation, new information regarding clinical studies related to NOTCH1 signaling inhibitors has been included in a new Section 2.1. named “NOTCH1 targeting therapeutic strategies” (page 4). This information is also summarized in new Figure 1 and Table 1.
- 6) Abbreviations should be clearly stated throughout the manuscript.
As suggested, abbreviations have been clearly stated throughout the manuscript.

Reviewer 3 Report
This is a well-written and comprehensive review, very well organized in describing the role of NOTCH signaling and its implications in T-ALL. There are only minor errors, which might be corrected later at the stage of proof.
However, it addresses and interestingly explores the complexity of Notch signaling in T-ALL. The manuscript is suitable for publication in its present form.
Author Response
Point by point to Reviewer 3 ‘s Comments
Comments and Suggestions for Authors
This is a well-written and comprehensive review, very well organized in describing the role of NOTCH signaling and its implications in T-ALL. There are only minor errors, which might be corrected later at the stage of proof.
However, it addresses and interestingly explores the complexity of Notch signaling in T-ALL. The manuscript is suitable for publication in its present form.
We thank Reviewer 3 for his/her comments. Minor errors have been corrected throughout the manuscript in the revised version that now accommodates concerns raised by Reviewers 1 and 2.

Round 2
Reviewer 2 Report
The manuscript can be considered in its present form for publication.